# The Development of a Screening Tool for Childcare Professionals to Detect and Refer Infant and Toddler Maltreatment and Trauma: A Tale of Four Countries

**DOI:** 10.3390/children10050858

**Published:** 2023-05-11

**Authors:** Elisa Bisagno, Alessia Cadamuro, Dierickx Serafine, Bou Mosleh Dima, Groenen Anne, Linde-Ozola Zane, Kandāte Annija, Varga-Sabján Dóra, Morva Dorottya, László Noémi, Rozsa Monika, Gruber Andrea, De Fazio Giovanna Laura, Blom Johanna Maria Catharina

**Affiliations:** 1Department of Biomedical, Metabolic and Neural Sciences, University of Modena and Reggio Emilia, 41121 Modena, Italy; 2Expertise Centre Resilient People, University Colleges Leuven-Limburg (UCLL), 3001 Leuven, Belgium; 3Department of Anthropology, Faculty of Humanities, University of Latvia, LV-1586 Riga, Latvia; 4Center Dardedze, LV-1002 Riga, Latvia; 5Pressley Ridge Hungary Foundation, 1142 Budapest, Hungary; 6Department of Law, University of Modena and Reggio Emilia, 41121 Modena, Italy; 7Center for Neuroscience and Neurotechnology, University of Modena and Reggio Emilia, 41121 Modena, Italy

**Keywords:** child abuse, maltreatment detection, 0–3 age group, three-layered screening tool, childcare professionals

## Abstract

Child maltreatment is considered a pressing social question, compromising the present and future mental and physical health of one in four children in Europe. While children younger than three years of age are especially vulnerable, few screening instruments are available for the detection of risk in this age group. The purpose of this research was to develop a screening tool for childcare professionals working in public and private daycare settings to support them in the early identification and referral of infants and toddlers exposed to emotional and physical abuse and neglect by primary caregivers, to be used in different settings across four European countries: Belgium, Italy, Latvia, and Hungary. Method: A stratified process was used to create the screening tool: We started by using Living lab methodology to co-create the screening tool with its final users, which was followed by testing the tool with a total of 120 childcare professionals from the four participating countries. Results: During the Living Lab phase, a screening tool with three layers was developed. The initial layer includes five “red flags” that signal particular concern and require immediate action. The second layer is a quick screener with twelve items focused on four areas: neglect of basic needs, delays in development, unusual behaviors, and interaction with caregivers. The third layer is an in-depth questionnaire that aids in formalizing a thorough observation of twenty-five items within the same four areas as the quick screener. After a one-day training session, 120 childcare professionals caring for children aged 0–3 from four countries assessed the screening tool and their overall training experience. Childcare professionals reported great satisfaction with the three-layered structure, which made the tool versatile, and agreed on its content, which was considered helpful in the daycare setting for the regular evaluation of the behavior of children and their primary caregivers, thus improving the early observation of change from the normal behavior of the infant or toddler. Conclusion: The three-layered screening tool was reported as feasible, practical, and with great content validity by childcare professionals working in four European countries.

## 1. Introduction

Each year, approximately one in four children and adolescents are exposed to child abuse and trauma or one or more adverse childhood experiences (ACEs) [1,2] (Figure 1). 

Regardless of the recent attention by policy efforts of the WHO and European Commission to reduce child abuse, children and adolescents continue to be victims and often face a high burden of present and future developmental problems, as well as frequently reduced physical, mental, social, behavioral, and emotional health [3]. 

The age group of 0–3 years old, which includes infants and toddlers, is the most vulnerable to abuse, with a high occurrence rate and the lowest likelihood of detection [4,5]. This makes it crucial to identify and screen for signs of child maltreatment and trauma. Failure to do so can have harmful consequences in the short and long term, affecting overall outcomes [5,6,7,8,9,10,11,12]. In this age group, maltreatment may lead to early-onset developmental disorders with profound negative outcomes, both at the behavioral level but also at the biological level, which tend to persevere and continue as adult conditions [13,14,15]. Consequently, the early recognition and identification of signs and symptoms will provide childcare professionals with a timely opportunity to intervene [16] and prevent the further worsening of extreme distress experienced by these very young children. 

The present reality is worrisome. Across Europe, fifty-five million children face abuse and neglect [3]. Thus, The WHO and The WHO European Regional Programme have implemented a new strategy to combat child maltreatment, shifting their focus from a protection-based approach, which relied on social and judicial methods, to a prevention-based approach that involves collaboration among various stakeholders in childcare across different domains [17,18]. To increase awareness of child maltreatment, the WHO European Regional Programme [3] urges member states to invest in (1) effective screening and (2) the formulation of national plans for preventing child maltreatment.

While widespread consensus [20,21] exists that the prevention of child abuse starts with the early recognition of signs, symptoms and risk factors, an enormous gap still exists between knowing, acknowledging and intervening [20], especially with regard to young children or professionals who are nearest to children (e.g., working in general settings such as educational and general wellbeing settings). At present, most screening tools for child abuse are developed for school-aged children three years and older, or healthcare specialists working in hospital settings [22,23]. However, in this context, checking on the child’s health occurs occasionally following country-specific newborn follow-up programmes on growth and development by professionals who see children infrequently. Childcare professionals, on the other hand, see infants and toddlers on a daily basis and are uniquely placed to monitor and observe the infant’s and toddler’s developmental trajectory [20]. They are thus among the first to encounter and observe primary caretaker abuse. Unfortunately, at present, there is a lack of accurate identification of the abuse of infants and toddlers in childcare settings, and there are no guidelines on when, how and what to screen for [20,23]. 

Most screening instruments are specific for healthcare settings and include medical history, the detections of signs using imaging instruments, subjective observations of child health and the interaction between child and caregiver. However, often, healthcare professionals simply rely on the statements and reports furnished by the primary caregiver, which, in 70% of cases, is the abuser [24]. Additionally, exposure to healthcare workers is not continuous and often limited to a few visits per year. However, in the healthcare setting, accurate and continuous assessment of early signs and symptoms improves detection substantially [25,26]. 

The majority of tests developed for infants and toddlers focus on physical symptoms [27], such as the TEN-4 assessing bruising [28,29]. Emotional abuse is considered inseparable from physical and sexual abuse and should be considered an integral part of any screening tool. One of the few tests developed for the healthcare setting that includes screening for behavioral and emotional symptoms is the INTOVIAN screening tool [30]. Other screening instruments specific for newborn home visits frequently also assess whether caregivers are at risk for abuse. Finally, little attention is given to neglect, the most prevalent form of abuse [23,24,31,32,33]. The identification of neglect requires the assessment of signs and symptoms belonging to different scopes, namely, (1) the physical scope of the infant/toddler (hygiene, clothing, overall health), (2) the scope of parenting difficulties or absence of resources to sustain the child in reaching its developmental milestones, and (3) other context-related scopes that place families under extreme distress (e.g., financial distress, illness, conflicts in the child environment, disability) [34]. Consequently, the lack of appropriate screening tools has caused personnel to rely strongly on intuition and experience, which is rather inaccurate [20,27]. Furthermore, because appropriate screening is lacking, the screening is often performed arbitrarily, when convenient for the professional or when it is absolutely clear that there is abuse.

As a result of the experience accumulated in the healthcare setting, evidence suggests that the appropriateness of a screening tool to detect the abuse of infants and toddlers varies according to the context in which it is used. Consensus exists that an efficient screening instrument should include the assessment of scientifically validated risk factors and the identification of physical, behavioral and emotional elements, which are especially important in preverbal infants and toddlers (0 to 3 years of age) [34,35]. 

Finally, the majority of child abuse is identified in the educational setting (at school) for children older than three years. Additionally, school-based detection and surveillance programs have been extremely successful in identifying and preventing future abuse in school-aged children by informing and educating both children and caregivers. Given this, we should put the same efforts into providing childcare professionals with the necessary means to detect and refer abuse of infants and toddlers by a primary caregiver [35,36].

As part of the research project “Enhancing the Capacity to combat child abuse through an Integral training and Protocol for childcare professionals” (ECLIPS), RECRDAP-GBV-AG-2020-101005642, the ECLIPS Screening and Referral protocol was developed to address some of the gaps mentioned earlier. This protocol is designed to offer childcare facility staff clear and concise guidance on how to: (a) detect signs and symptoms of trauma, (b) report suspected or known child abuse to the authorities and (c) communicate with primary caregivers, such as parents, about their observations. The complete protocol can be found on the ECLIPS website (https://eclipsproject.eu/, accessed on 3 May 2023) in English, Italian, Latvian, Dutch and Hungarian. The protocol comprises different tools, namely: (a) detailed information on infants’ and toddlers’ typical developmental stages and milestones, as well as a description of the most frequent signs and symptoms of child abuse; (b) a self-assessment tool for childcare professionals’ wellbeing; (c) the three-layered screening tool; (d) referral guidelines for childcare professionals to refer suspects of child abuse; and (e) detailed guidelines and tools to support communication with primary caregivers. For the purpose of this paper, the screening tool content, development and assessment will be detailed.

## 2. Materials and Methods

### 2.1. Screening Protocol Development: Design and Procedure

The ECLIPS screening tool was designed as a four-phase process. Here, we report on phases 3 and 4. The main outcome of phases 1 and 2, which was designed to identify the needs of childcare professionals regarding the screening and referral process for child abuse, was a general lack of tools, expertise and training considering screening, as well as the need for referral guidelines. Detailed information can be found in Bisagno et al., 2023 [20]. Based on the first two phases, in Phase 3 we engaged an expert panel of key stakeholders in a co-creation process using a Living lab methodology to develop the domains, format and structure, and to be included in the screening tool [37]. Phase 4 involved training various stakeholders in the use of the tool, followed by a comprehensive assessment that involved 120 childcare professionals from the four partner countries (namely, Belgium, Hungary, Italy and Latvia) who performed an evaluation of the positive aspects and usability as well as the challenges posed by the tool (Figure 2, see Step 3 and 4).

#### 2.1.1. The Living Labs

Two out of the four partners took the lead on the screening protocol (while the other two designed a trauma-informed care protocol, i.e., authors, under review), namely an academic partner, the University of Modena and Reggio Emilia (Unimore: Italy), and an NGO partner, Centrs Dardedze (Latvia). Both Unimore and Centrs Dardedze arranged Living Labs (LLs) to collect ideas, feedback and to co-create the tool. An LL is defined as “a research methodology for sensing, prototyping, validating and refining complex solutions in multiple and evolving real-life contexts” [37]. A specific characteristic of LLs is that the final users of a product are considered its “co-producers” and take an active part in designing it [38].

Both in Italy and Latvia, ten to fifteen participants took part in each of the eight LLs. Participants were practitioners, namely, childcare professionals, managers of childcare facilities, as well as researchers (pediatrics, developmental psychology, and/or criminology), and at least two parents, who offered a complementary vision regarding the need for screening in childcare facilities. The purpose of the LLs was to provide input regarding the perceived needs related to screening and referral and, based on these, to co-create an “ideal” screening tool for childcare professionals.

Unimore and Centrs Dardedze organized eight intranational LLs each between May 2021 and February 2022, suitably interspersed with meetings between project partners to proceed with a homogeneous structure. The content of the eight LLs was the following:*LL meetings 1 and 2, exploration stage*: assessment needed concerning screening, discussion, and brainstorming about results of a previous inventory of the good practices and the insights from expert focus groups [20].*LL meetings 3–6, co-construction stage*: joint development of the screening tool*LL meetings 7–8, piloting stage*: meetings 7 and 8 took place one month after one another, during which the LLs’ childcare professionals practiced the use of the screening tool within their facilities and shared it with colleagues. Based on their experiences, both professionals and parents offered feedback on the LLs. Modifications were implemented by the researchers and led to the finalizing of the protocol.

The final form of the screening protocol was then defined by Unimore and Centrs Dardedze and reviewed by external experts (e.g., Italian and Latvian psychologists, policymakers and other stakeholders) as well as by the other two ECLIPS partners (the Belgian UC Leuven-Limburg and the Hungarian NGO Pressley Ridge) until it assumed the final form.

#### 2.1.2. Education and Testing

A training process was developed with the purpose of testing the screening tool. In all four project partner countries, one-day training with two groups of 10–20 childcare professionals was held. The participants were recruited on a voluntary basis via advertising. The trainers were the project partners’ researchers and professionals, together with at least one childcare professional from the LLs. Within the training, participants were presented with theoretical knowledge regarding the phenomenon of child abuse within the 0–3 age group, as well as the importance of screening. They were also shown the screening tool, which was presented and explained in every detail. Some practical simulations were used to allow a first approach to the tool. After the one-day learning group training, participants were each provided with a copy of the screening tool, with the task to keep testing it weekly for at least one month.

### 2.2. Measures and Data Analysis

The participants’ knowledge of child abuse signs and symptoms (e.g., “Trauma symptoms are visible in babies and toddlers”) and their attitude towards the role of childcare professionals in screening and referring child abuse (“Childcare professionals can play an important role in screening and referring child abuse”, single item) were tested via an ad hoc anonymous survey administered three times (i.e., T0 = right before the training; T1 = one month after the training; T2 = three months after the training). Participants’ evaluation of the screening tool (e.g., “I found the Screening Tool to be useful”) and of the training as a whole (e.g., “The content of the program aligned with my expectations”) was recorded twice, at T1 and T2. Responses were recorded on a 5-point Likert scale, with options ranging from 1 (Strongly disagree) to 5 (Strongly agree), where higher scores indicated a more favorable evaluation.

To respond to the specific needs of the training evaluation, the questionnaire we used was based on different existing measures. Specifically, the first seven items were adapted from the Knowledge, Attitudes and Practices of Trauma Informed Practice survey created by Abdoh et al. (2017) [39] for a healthcare professional population. This tool is a self-report measure of perceived knowledge, attitudes and practical actions that healthcare professionals declare with respect to trauma-informed care. We adapted the items to childcare professionals and to the topic of the screening and referral of child abuse. The training evaluation items were instead derived and adapted from Verdonck et al. (2019) [40]. A first draft of the questionnaire was presented to the ECLIPS consortium, and each partner presented their notes. This feedback was then incorporated into the final version. The whole questionnaire is presented in Appendix A.

Mean scores were calculated for the knowledge, screening tool appreciation and training appreciation components, while the attitude component was measured via a single item. Means and *SDs* for each score were calculated at different measurement times. A series of two-way ANOVAs (time × country) were used to calculate differences from T0 to T2 in the four countries.

## 3. Results

### 3.1. Outcome Co-Creation Process

The screening tool is tailored specifically for infants and toddlers between the ages of 0 and 3 years (the full ECLIPS Screening and Referral protocol can be found here: https://eclipsproject.eu/wp-content/uploads/2022/04/eclips-screening-and-referral-protocol-screen.pdf, accessed on 3 May 2023). The protocol is available in English, Italian, Latvian, Dutch and Hungarian. Based on scientific literature and childcare professionals’ observations of the child’s appearance, behavior and interactions with caregivers, the tool includes lists of signs and symptoms that childcare professionals can use to perform an (ideally) monthly screening of infants and toddlers in the childcare setting. A critical request from childcare professionals during the development of the tool was for it to be detailed yet quick to fill out, to match their busy schedules. Therefore, the screening tool has been designed with three layers, namely a “red flag” list, a quick screener and an in-depth questionnaire (which is only meant to be filled out in the case of one of the two previous layers reaching a specific threshold). It is not advisable to use the screening tool during the child’s initial entry into the daycare environment, typically lasting two to four weeks, as certain indicators listed in the tool may be attributed to the child’s adjustment period rather than potential abuse. The structure of each layer is detailed below.

#### 3.1.1. First Layer: The Five “Red Flags”

The “red flags” refer to five specific signs that are of significant concern and require an immediate response. If any one of these red flags is identified, it is mandatory for the professional to proceed with the “in-depth questionnaire” and concomitantly follow the so-called “red action mode” for referral (which requires immediate referral to the competent authorities), irrespective of the outcome of the in-depth screening. This is because the red flags are per se worrying enough to proceed to referral; however, the “in-depth questionnaire” is meant to gather more precise information to specify the observed signs and symptoms, and therefore inform the referral process. An example of an item in the “red flag” layer is “The caregiver is verbally and/or physically aggressive towards the child in the presence of the childcare specialist and/or another primary caregiver”. The complete list of items can be found on page 23 of the English screening tool.

#### 3.1.2. Second Layer: The “Quick Screener”

The quick screener is a list of 12 signs and symptoms, for which the childcare professional answers in a dichotomous way (yes/no) if they have occurred during the last month. The signs and symptoms regard four areas, which were defined both based on clinical developmental psychology literature and on those of other instruments. The areas are: (a) neglect of basic needs (e.g., “The child is often dressed in dirty clothes and/or is unclean/unhygienic for a month”); (b) delays in development (e.g., “The child does not reach the expected developmental milestones”); (c) unusual behaviors (e.g., “The child hurts themself”); and (d) interaction with the caregivers (e.g., “The child “freezes” at the sight of a caregiver or other adult”). If at least three of these 12 signs are identified by the childcare professional, the child is subjected to further in-depth screening. This cut-off, which still needs to be tested, was agreed on with the participants of the LLs as a number of signs relevant enough to be worrying. The complete list of items can be found on page 24 of the screening tool.

#### 3.1.3. Third Layer: The “In-Depth Questionnaire”

These are 25 signs, with a higher degree of detail but divided into the same four areas as the quick screener, namely, (a) neglect of basic needs (e.g., “The caregiver forgets to pick up the child, arrives very late or the child is absent from the childcare facility for no justified reasons”); (b) delays in development (e.g., “The child tends NOT to move, crawl or walk”); (c) unusual behaviors (e.g., “The child rapidly alternates between emotions and moods without apparent reason”); and (d) interaction with the caregivers (e.g., “The child actively avoids contact with the primary caregiver by moving away, crying, appearing frightened, or stiffening in the presence of the caregiver”). The professional assesses these items on a 4-point Likert frequency scale ranging from 0 (*Never*) to 3 (*Frequently*) to obtain a score. The scoring tool utilizes the traffic light principle for grading, with a green score indicating no cause for concern (scores below 31), and an amber score signaling a potential risk of violence (scores between 32 and 49), where advice from specialized agencies, such as Social Services, should be sought. A red score (scores of 50 or higher) denotes that an urgent referral is necessary. Similar to those of the “Quick screener”, these cut-offs were also agreed upon by the LLs participants but require further testing. Each of the colors of the traffic light guides a corresponding “action mode”. The complete list of items can be found on pages 25–26 of the screening tool.

#### 3.1.4. Referral Flowchart

Based on the traffic light principle (green–amber–red), a certain “action mode” is then activated. In the “green action mode”, no worrying signs are detected. Childcare professionals should simply continue the routine screening, and potentially inform parents of the results, if they wish. In the “amber action mode”, childcare professionals should inform their supervisors of the result and ideally consult with appropriate authorities (Social Services); another suggested action would be to organize a meeting with the child’s parents to discuss the identified risk and take care to repeat the screening process after a month. Lastly, in the “red action mode”, childcare professionals are required (by law, in most European countries) to immediately report to their supervisor and possibly to law enforcement. It should be noted that while the flowchart is intended as a general guide for all European countries, it is crucial to take into account the unique judiciary systems of each country, region, and/or territory to determine the most suitable institution to refer to, based on the resulting “action mode”.

### 3.2. Outcome Training and Testing

A total of 100 childcare professionals took part in the training within the four countries and filled out the questionnaire at T0, with a 26% attrition rate at T2 (see Table 1). No personal data variables were collected to guarantee participants the most complete anonymity in completing the questionnaire. The questionnaire was completed in the mother tongue of the participants.

Table 2 presents the *Means* and *SDs* for all variables at the different measurement times. To detect a potential modification in childcare professionals’ knowledge of child abuse signs and symptoms (“Knowledge”), attitude towards the role of childcare professionals in screening and referring child abuse (“Attitude”), as well as the evaluation of both the screening tool (“Screening”) and the training as a whole (“Training”), a series of 3 × 4 ANOVAs were performed with time (T0, T1, T2) and country (Belgium, Italy, Hungary, Latvia) included as factors. It is worth noting that for screening and training, no data were collected at T0 (before the training).

A two-way ANOVA was executed to analyze the effect of time and country on self-reported knowledge. The analysis revealed that there was a statistically significant interaction between the effects of time and country (*F*(6, 273) = 2.693, *p* = 0.015). Simple main effects showed that both time (*F*(2, 273) = 82.466, *p* < 0.001) and country (*F*(3, 273) = 38.646, *p* < 0.001) had significant effects on self-reported knowledge.

As can be noticed in Figure 3, all childcare professionals increased in self-reported knowledge, with the highest increase in Italian childcare professionals, especially between T0 (*M* = 3.011) and T1 (*M* = 3.833). A similar pattern was found in Latvian childcare professionals, who showed a considerable increase in self-reported knowledge between T0 (*M* = 3.763) and T1 (*M* = 4.332), followed by a plateau between T1 and T2 (*M* = 4.462). Belgian participants showed a more linear increase from T0 (*M* = 3.588) to T2 (*M* = 4.235), similar to Hungarian participants (*M*_T0_ = 3.957; *M*_T2_ = 4.616).

The ANOVA on self-reported attitude showed no significant interaction between the effects of time and country (*F*(6, 273) = 1.126, *p* = ns). However, both time (*F*(2, 273) = 5.317, *p* = 0.005) and country (*F*(3, 273) = 3.948, *p* = 0.009) had significant main effects on self-reported attitude (see Figure 4). Specifically, Hungarian participants consistently showed the most positive attitude: on T0, their mean score was 4.850 compared to one significantly lower for the other countries (*M*_Belgium_ = 4.590; *M*_Italy_ = 4.700; *M*_Latvia_ = 4.460). The same is true at T2, with Hungarian participants showing a ceiling effect on attitude (*M* = 5.000) and the other countries scoring 0.17 to 0.27 lower (*M*_Belgium_ = 4.820; *M*_Italy_ = 4.830; *M*_Latvia_ = 4.730). Both Italian and Latvian childcare professionals showed an improved attitude from T0 to T1, followed by a plateau. Belgian participants showed a slight downturn between T0 (*M* = 4.460) and T1 (*M* = 4.370), followed by a significant increase at T2 (*M* = 4.730). Most importantly for the present article, the ANOVA on the evaluation of the screening tool (“Screening”) showed no significant interaction between the effects of time and country (*F*(6, 273) = 1.875, *p* = ns), as well as no main effect of a country (*F*(3, 273) = 1.748, *p* = ns). Time (*F*(2, 273) = 12.217, *p* = 0.001), however, caused a significant increase in the perception of the usefulness of the screening tool in all countries, with the highest increase from T1 (*M* = 3.833) to T2 (*M* = 4.500) shown by Belgium (Figure 5).

Lastly, considering the general evaluation of the training, the analysis revealed that there was a statistically significant interaction between the effects of time and country (*F*(6, 273) = 3.343, *p* = 0.021). Simple main effects showed that both time (*F*(2, 273) = 4.334, *p* = 0.039) and country (*F*(3, 273) = 7.694, *p* < 0.001) had significant effects on training. All countries displayed a very positive initial evaluation of the training (all scores over 4.00). However, while Italian and Latvian participants showed a relatively stable evaluation of the training, with Δs ranging from −0.12 to +0.01 (*p* = ns), Belgian and Hungarian participants showed a consistent increase in their evaluation of the training between T1 (M_Belgium_ = 4.125; M_Hungary_ = 4.420) and T2 (M_Belgium_ = 4.422; M_Hungary_ = 4.815) (Figure 6).

## 4. Discussion

Childcare professionals working in daycare and home visiting settings occupy a privileged position of being able to detect the risk or presence of caretaker abuse in a timely fashion because of their close, regular and often prolonged daily contact with infants and toddlers; however, they often lack the tools and training to do so [20]. Our project had a twofold purpose: the primary purpose was to develop a screening tool with and for childcare professionals that is easy to use, includes different areas of observation (physical, emotional, behavioral, caregiver interaction) [13,41,42,43,44,45] and is accompanied by a risk-based assessment tree that facilitates the decision-making process in case of the risk or presence of child abuse [46,47]. Secondly, a training plan was developed to accompany the childcare professional in acquiring the ability to feel comfortable using the tool.

We have also discovered that involving childcare experts from the beginning is crucial for developing a tool that is responsive to developmental changes and serves as a reliable screener for promptly detecting infants and toddlers who may be at risk of or exhibiting signs of abuse. The Living Lab approach, which involves the end user in the product development process, has proven to be the optimal solution for creating this tool. Drawing on our experience with developing the ECLIPS screening tool, however, we believe that the LL methodology could benefit from inserting a degree of flexibility. This might involve adding extra meetings as needed, as well as gradually implementing the tool in practical contexts during its development. By doing so, we can gather timely and factual feedback, in addition to the subjective evaluations of the LL participants. Finally, given that all screening tools require appropriate and adequate professional knowledge and training for their use and the documentation of the collected data, the ECLIPS three-step screening tool fills this gap. Moreover, the tool also includes assistance and education to improve the communication and relationship-building skills of parents, which is commonly lacking while considered fundamental [48,49].

Data from the childcare assessment showed that all childcare professionals, after following the training and experimenting with the screening tool for one month, increased in self-reported knowledge of child abuse signs and symptoms, showing that both the training and access to the tool helped them learn more about children’s milestones in development and signs of abuse. Moreover, it is worth noting that childcare professionals also improved overall in terms of attitude, meaning that they reported higher self-efficacy related to the importance of their role in detecting signs and symptoms of abuse in infants and toddlers. Importantly, all childcare professionals positively evaluated the screening tool immediately after the training, and evaluated it as even better after using it for two months, suggesting that, with practice, it becomes easier to use and they also could explore and appreciate the opportunities it offers.

The ANOVAs revealed interesting differences among the four participating countries regarding their knowledge of screening and referral, attitudes and evaluation of the ECLIPS screening tool. Hungarian participants had the highest knowledge and most positive attitude from the start, possibly due to the specialized structure of childcare services in Hungary. For Italy and Latvia, where childcare professionals typically do not receive specific training on child abuse during their education, the training led to a significant increase in knowledge and attitude, particularly in Latvia where a previous study showed a knowledge gap and dismissive attitude. The ECLIPS screening tool’s greatest strengths lie in the positive evaluation it has received from childcare professionals, its collaborative design process with end users, and its status as the first tool specifically developed for the educational community to identify signs and symptoms of child abuse in the 0–3 age group.

Although the tool has undeniable qualities, it has limitations and room for improvement. One important aspect to consider is ensuring the tool’s accuracy by providing adequate training. This includes information on child development, child abuse prevalence, and practical guidelines on conducting observations, calculating scores, and making referrals based on national context. This is especially important for countries where childcare professionals do not receive specific training on child abuse during their standard training. To address this limitation, an online training course could be developed to ensure the widespread and effective use of the screening tool. An important challenge for the future is to develop asynchronous training programs to make the ECLIPS screening tool accessible to as many childcare professionals as possible. Secondly, the ECLIPS tool does not currently represent an accurate differentiation with standardized cutoffs based on population norms. This limitation does not allow the use of the tool within clinical contexts. Future studies should investigate the psychometric properties of the tool with the specific intent of establishing standardized cutoffs for the clinical detection of cases of abuse.

While the potential of our screening protocol to assist childcare professionals working in private or public daycare is an important one, it is time for us, as a larger community, to assume shared responsibility, bringer greater trust and transparency to the screening of child abuse in infants and toddlers who cannot express themselves in an articulate verbal manner.

Dedication is not the rate-limiting factor in adopting the screening tool, but trust in one’s own role, capacity and time are. Therefore, if we truly want to turn the page, a shift in policy is necessary. All childcare, and especially the care of very young children, revolves around expectations regarding the professionality of the childcare professional.

Childcare professionals are expected to handle sensitive issues concerning vulnerable individuals such as children and their caregivers. The ECLIPS screening tool provides only a part of the solution, and the development of adequate training programs that engage professionals in the design process is necessary. This co-creation approach leads to critical appraisal, a willingness to be involved, and formal recognition of commitment. While screening tools like the three-layered screening and referral tool provide immediate help, educational programs are needed to increase the technical content-related literacy of all stakeholders [50,51]. The development of an easy-to-use tool for screening and referring child abuse in infants and toddlers specifically for childcare professionals is a much-needed innovation. Overregulation and overburdening of professionals should be avoided, and useful tools and policies should be embraced to a prevent lack of training and regulatory neglect. Ultimately, this practical screening and referral tool can accelerate innovation while ensuring transparent and trustworthy progress towards better protection of the most vulnerable [52].

## Figures and Tables

**Figure 1 children-10-00858-f001:**
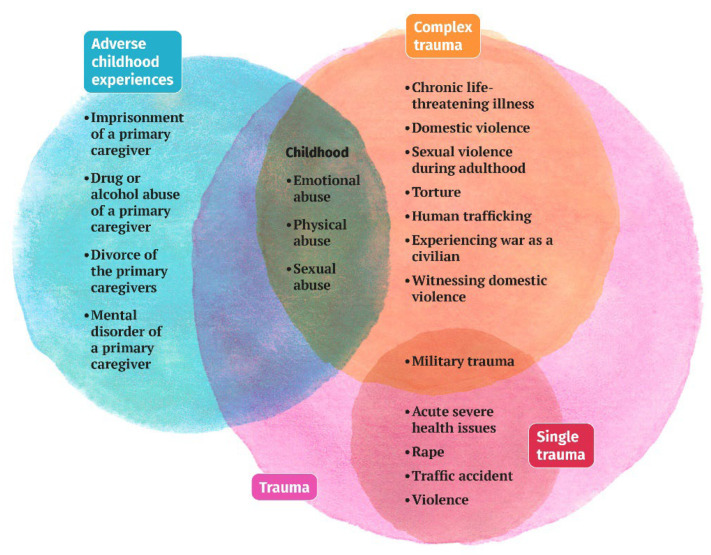
Clinical presentation of ACEs in children aged 0–3. Note. Adapted from the National Health Service Education for Scotland [19]. Image by Emese Iványi.

**Figure 2 children-10-00858-f002:**
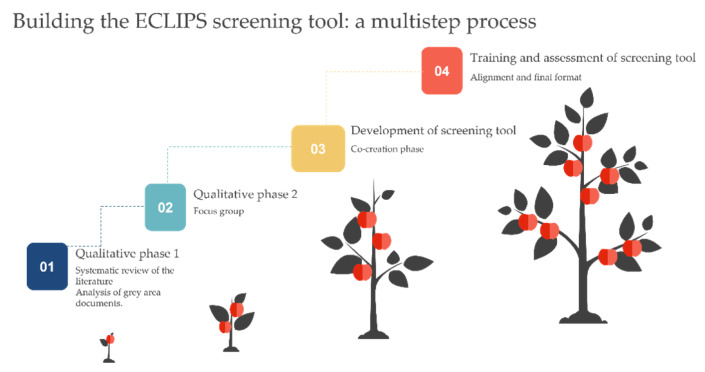
Procedure and development of the ECLIPS screening tool. Phases 1 and 2 have been explained elsewhere [20]. Phases 3 and 4, respectively, represent the co-creation phase and the training and assessment phase.

**Figure 3 children-10-00858-f003:**
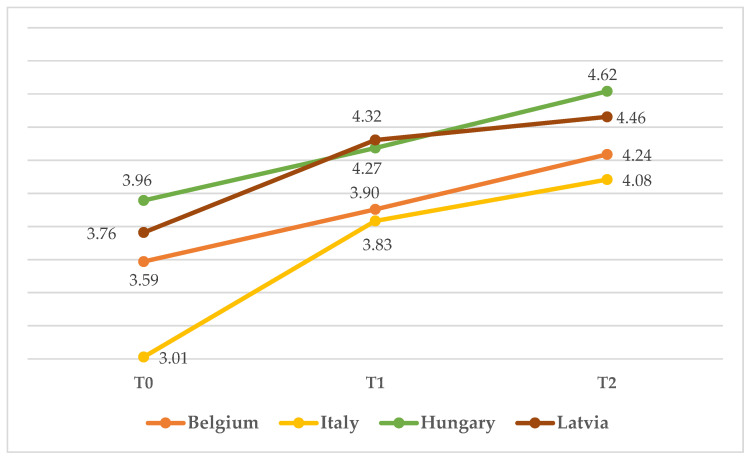
Mean score on knowledge from T0 to T2 in the four countries.

**Figure 4 children-10-00858-f004:**
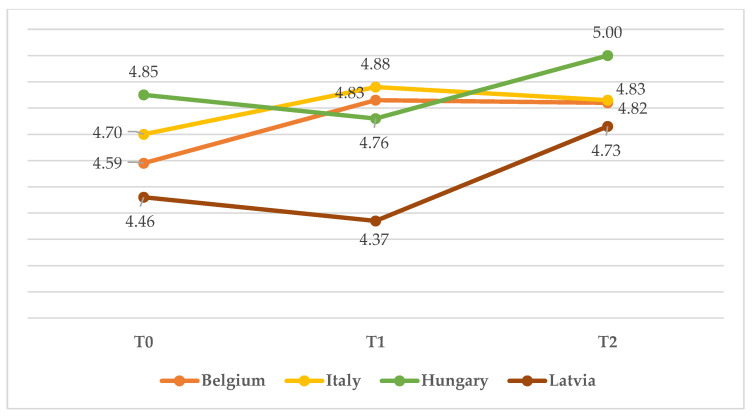
Mean score on attitude from T0 to T2 in the four countries.

**Figure 5 children-10-00858-f005:**
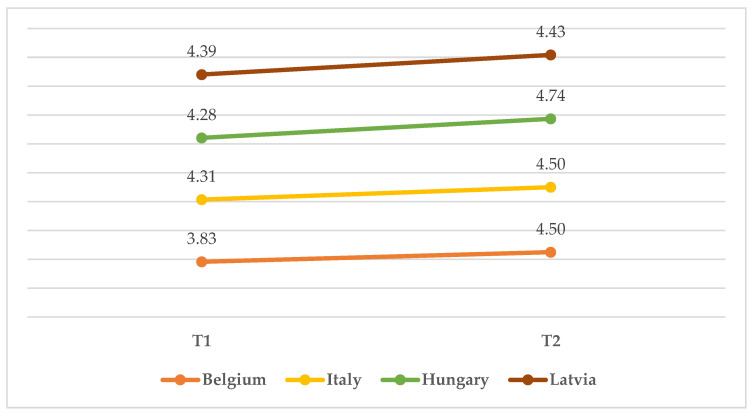
Mean score on screening from T1 to T2 in the four countries.

**Figure 6 children-10-00858-f006:**
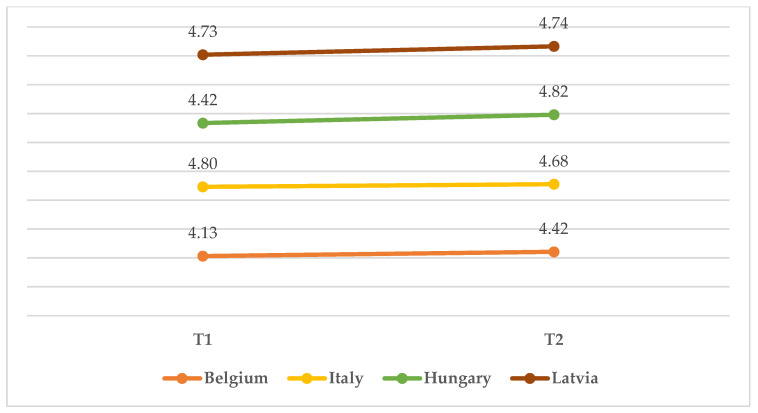
Mean score on training from T1 to T2 in the four countries.

**Table 1 children-10-00858-t001:** Overview of participating childcare professionals.

PP	T0	T1	T2
UCLL (Belgium)	17	12	17
UNIMORE (Italy)	30	26	30
Pressley Ridge (Hungary)	27	27	27
Centrs Dardedze (Latvia)	26	30	22
Total sample	100	95	74

Note. PP = project partner; T = measurement time.

**Table 2 children-10-00858-t002:** Means and standard deviations of each variable divided per country and measurement time.

Country	Time	Knowledge		Attitude		Screening		Training	
		*M*	*SD*	*M*	*SD*	*M*	*SD*	*M*	*SD*
**Belgium**	**T0**	3.59	0.37	4.59	0.51	-	-	-	-
	**T1**	3.90	0.39	4.83	0.39	3.83	1.14	4.13	0.58
	**T2**	4.24	0.24	4.82	0.39	4.50	0.54	4.423	0.38
**Italy**	**T0**	3.01	0.56	4.70	0.47	-	-	-	-
	**T1**	3.83	0.37	4.88	0.33	4.31	0.60	4.80	0.35
	**T2**	4.08	0.37	4.83	0.38	4.50	0.58	4.68	0.53
**Hungary**	**T0**	3.96	0.44	4.85	0.36	-	-	-	-
	**T1**	4.27	0.44	4.76	0.53	4.28	0.61	4.42	0.56
	**T2**	4.62	0.29	5.00	0.00	4.74	0.60	4.82	0.36
**Latvia**	**T0**	3.76	0.51	4.46	0.51	-	-	-	-
	**T1**	4.32	0.44	4.37	0.45	4.39	0.51	4.73	0.40
	**T2**	4.46	0.30	4.73	0.55	4.43	0.59	4.74	0.44

## Data Availability

The data that support the findings of this study are available from the corresponding author, J.M.C.B., upon reasonable request.

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
