# Peer review of "The Development of a Screening Tool for Childcare Professionals to Detect and Refer Infant and Toddler Maltreatment and Trauma: A Tale of Four Countries"

_children, 2023, doi:10.3390/children10050858_

Round 1

Reviewer 1 Report

The creation of a screening tool for maltreatment for childcare professionals working in public and private daycare settings with children up to age 3 is important given that few such instruments exist as stated by the authors.   Having such a tool at their disposal provides childcare professionals a systematic approach with specific criteria to informing screening and assessment that is important not only in practice but also in enhancing the protection and safety of young children.  It is important to understand the process of how such instruments are developed and how that process as well as the products from that process are received and experienced. For these reasons, this study is interesting. 

The introduction and methods explain the purpose of the study and research activities undertaken to execute the study design, evaluate/measure training, collect and analyze data.

Can the authors provide additional details about the training Evaluation tool ? its reliability? face validity? How were the items selected (from the literature? developed by the authors? are from an existing measure?). 

The tool is a self-report and included items that measure  perceived knowledge as opposed to being an actual knowledge test ? Is that correct? 

The links provided to ECLIPS do not work/not active.  Readers most likely will be interested in examining  instrument as it is being described in the narrative.

The results section can be improved.  In the results section, sections 3.1 to 3.1.4, the authors provide a description of the layers of the screening tool that resulted from co-creation process and Living Labs (LL). In addition to the products that were developed through these activities and described here,  this section can be enhanced by the authors providing any data (quantitative/qualitative) that speak to the participants evaluation of their experiences with the co-creative process and Living Labs ( positive, negative, neutral, strengths/limitations of using these approaches) as well as their assessment of each layer of the tool, beyond overall usefulness and user-friendly aspects. 

Figures 3, 4, 5, 6 - the  bar/line colors representing each country should be more strongly contrasted so that they are more easily distinguishable from each other (alternatively, each bar/line in the body of the figure could be labeled with the corresponding country). 

The results indicated significant differences between countries when contrasted on knowledge, attitudes, etc. What are the implications of these country to country differences for cross-country and cross-cultural development of screeners and training to use those screeners? 

The discussion section can be improved. The authors should comment on the strengths and limitations of the evaluation method used that may have contributed to the results found and reported.

What recommendations/lessons learned do the authors want to pass along to other researchers who may be interested in using the co-creation process given experiences with the co-creation process and training evaluation findings? 

Author Response

Hereby, I’m enclosing the revised version of the manuscript children-2340666, entitled “The development of a screening tool for childcare professionals to detect and refer infant and toddler maltreatment and trauma: A tale of four countries” to Children. Please find attached the point-by-point response to all comments made by the reviewers. As suggested, the changes and additions are highlighted in track change in the paper. In blu the issue raised by the reviewers in black our reply. The line numbers refer to the clean version. We rewrote the parts indicated containing too much repletion. However, please note that the hyphened parts cannot be changes as the are the literal wording of items on the questionnaire.

We think the manuscript has improved with the input of the reviewers and hope that you consider it now fit for acceptance in your journal.

Sincerely,

Johanna Blom

Detailed reply to Reviewer: 1

The creation of a screening tool for maltreatment for childcare professionals working in public and private daycare settings with children up to age 3 is important given that few such instruments exist as stated by the authors. Having such a tool at their disposal provides childcare professionals a systematic approach with specific criteria to informing screening and assessment that is important not only in practice but also in enhancing the protection and safety of young children.  It is important to understand the process of how such instruments are developed and how that process as well as the products from that process are received and experienced. For these reasons, this study is interesting.

We thank the Reviewer for their positive evaluation. We also believe that a tool like this is very much needed by childcare professionals and we hope that we’ll have the possibility to keep improving it in the near future.

The introduction and methods explain the purpose of the study and research activities undertaken to execute the study design, evaluate/measure training, collect, and analyze data. Can the authors provide additional details about the training evaluation tool? Its reliability? Face validity? How were the items selected (from the literature? developed by the authors? are from an existing measure?).

Thank you for the suggestion. We added as many details as possible concerning this topic. The reliability and face validity of the tools we adapted are not reported in the original papers; however, we detailed the process of the items adaptation and revision by the ECLIPS consortium (see lines 229-237).

The tool is a self-report and included items that measure perceived knowledge as opposed to being an actual knowledge test? Is that correct?

That is correct, and it is based on the tool we used as an adaptation. We clarify this in lines 229-237.

The links provided to ECLIPS do not work/not active. Readers most likely will be interested in examining instrument as it is being described in the narrative.

Thank you for bringing this up. We checked all of the links and they seem well-functioning. We’ll check again if this problem seems to persist for other users.

The results section can be improved.  In the results section, sections 3.1 to 3.1.4, the authors provide a description of the layers of the screening tool that resulted from co-creation process and Living Labs (LL). In addition to the products that were developed through these activities and described here,  this section can be enhanced by the authors providing any data (quantitative/qualitative) that speak to the participants evaluation of their experiences with the co-creative process and Living Labs (positive, negative, neutral, strengths/limitations of using these approaches) as well as their assessment of each layer of the tool, beyond overall usefulness and user-friendly aspects.

Unfortunately, we did not collect such data about the LL process because we used it as a (already validated) tool aimed at a scope (which, for us, was developing the Screening and Referral tool) rather than a process to be evaluated. Possible informal feedback that we can give about the LL process is that all stakeholders, in the end, were proud of the product and the childcare professionals immediately started to use it within their childcare facilities.

Figures 3, 4, 5, 6 - the  bar/line colors representing each country should be more strongly contrasted so that they are more easily distinguishable from each other (alternatively, each bar/line in the body of the figure could be labeled with the corresponding country).

We now use different colors to better distinguish the lines – if they will be changed according to the journal’s guidelines we’ll substitute the colors with a label as suggested.

The results indicated significant differences between countries when contrasted on knowledge, attitudes, etc. What are the implications of these country to country differences for cross-country and cross-cultural development of screeners and training to use those screeners?

We thank the Reviewer for the question. We provided our answer to this in lines 429-440.

The discussion section can be improved. The authors should comment on the strengths and limitations of the evaluation method used that may have contributed to the results found and reported.

Thank you for pointing this out. Based on both R1 and R2’s suggestions we stressed and better detailed strenghts and limitations of the paper (see lines 439-469) and offered a general conclusion (see lines 474-479).

What recommendations/lessons learned do the authors want to pass along to other researchers who may be interested in using the co-creation process given experiences with the co-creation process and training evaluation findings?

We added some information with respect to this at lines 469-473.

Reviewer 2 Report

Interesting work from a practical point of view. I congratulate the authors on the concept and the effort made. In the manuscript presented for review, I miss the Limitations chapter, which defines the basic limitations associated with the study.

Please improve the conclusions so that they are more general in nature. Conclusions must be one or more sentences that coherently and unambiguously summarize the results obtained and provide a generalization.

Author Response

Hereby, I’m enclosing the revised version of the manuscript children-2340666, entitled “The development of a screening tool for childcare professionals to detect and refer infant and toddler maltreatment and trauma: A tale of four countries” to Children. Please find attached the point-by-point response to all comments made by the reviewers. As suggested, the changes and additions are highlighted in track change in the paper. In blu the issue raised by the reviewers in black our reply. The line numbers refer to the clean version. We rewrote the parts indicated containing too much repletion. However, please note that the hyphened parts cannot be changes as the are the literal wording of items on the questionnaire.

We think the manuscript has improved with the input of the reviewers and hope that you consider it now fit for acceptance in your journal.

Sincerely,

Johanna Blom

Detailed reply to Reviewer: 2

Interesting work from a practical point of view. I congratulate the authors on the concept and the effort made. In the manuscript presented for review, I miss the Limitations chapter, which defines the basic limitations associated with the study.

We thank the Reviewer for their positive feedback and comments that helped improving the final version of this manuscript. We stressed and better detailed strengths and limitations of the paper (see lines 439-469).

Please improve the conclusions so that they are more general in nature. Conclusions must be one or more sentences that coherently and unambiguously summarize the results obtained and provide a generalization.

We thank the Reviewer for this suggestion. We did as suggested, see lines 472-478, and especially the last line (476-479).
